# Zigzag Fetal Heart Rate Pattern in an Uncomplicated Pregnancy with Dual Intrauterine Infection Detected During Labor with Intact Membranes: A Case Report

**DOI:** 10.3390/healthcare13141726

**Published:** 2025-07-17

**Authors:** Martina Derme, Valentina Demarco, Adele Vasta, Paola Galoppi, Ilenia Mappa, Giuseppe Rizzo

**Affiliations:** Department of Maternal and Child Health and Urological Sciences, Policlinico Umberto 1, Università Roma Sapienza, 00189 Rome, Italy; martina.derme@uniroma1.it (M.D.); valentina.demarco@uniroma1.it (V.D.); adele.vasta@uniroma1.it (A.V.); paola.galoppi@uniroma1.it (P.G.); mappa.ile@gmail.com (I.M.)

**Keywords:** histological chorioamnionitis, subclinical chorioamnionitis, intrauterine infection, zigzag pattern, reduced fetal movements

## Abstract

**Background**: Histologic chorioamnionitis (HCA) is a placental inflammatory condition characterized by neutrophilic infiltration of the fetal membranes, often occurring without overt clinical signs or symptoms. Risk factors include prolonged labor, premature rupture of membranes (PROM) exceeding 12 h, nulliparity, labor dystocia, and lower socioeconomic status. Although HCA frequently presents as a subclinical condition, its early diagnosis remains challenging. Nevertheless, HCA is associated with an increased risk of maternal and neonatal morbidity, including early-onset neonatal sepsis, cerebral palsy, and long-term neurodevelopmental impairment. We report the case of a 29-year-old primigravida at 40 + 0 weeks of gestation, admitted for decreased fetal movements. **Discussion**: Cardiotocographic (CTG) monitoring revealed a “zigzag pattern” in the absence of maternal fever, leukocytosis, or tachycardia. Due to the CTG findings suggestive of possible fetal compromise, in addition to reduced fetal movements, an emergency cesarean section was performed. Intraoperative findings included heavily meconium-stained amniotic fluid, then the examination of the placenta confirmed acute HCA with a maternal inflammatory response, without evidence of fetal inflammatory response. **Conclusion**: This case highlights the crucial role of CTG abnormalities, particularly the “zigzag pattern,” as an early marker of subclinical intrauterine inflammation. Early recognition of such patterns may facilitate timely intervention and improve perinatal outcomes in cases of histologic chorioamnionitis.

## 1. Introduction

Chorioamnionitis is an infection and inflammation of the chorion, amnion, or both, as well as of the fetus and umbilical cord. It may be associated with potentially serious maternal and fetal consequences and is considered the most important cause of both maternal and neonatal sepsis [1,2,3].

According to the reVITALize Initiative, chorioamnionitis is defined by the presence of unexplained fever (>38 °C or 100.4 °F) along with one or more of the following signs or symptoms: uterine tenderness and irritability, leukocytosis, fetal tachycardia, maternal tachycardia, and malodorous vaginal discharge [4].

It is important to distinguish between ‘clinical chorioamnionitis’ (CCA) and ‘histologic chorioamnionitis’ (HCA). HCA is a pathological diagnosis, defined by polymorphonuclear leukocyte infiltration, indicating placental inflammation and necrosis. Although CCA and HCA are related, HCA can occur without clinical symptoms [2,5].

The distinction between clinical and histologic chorioamnionitis is relevant not only diagnostically but also from a public health perspective. While clinical cases may prompt immediate intervention, histologic forms often go unrecognized until after delivery, limiting timely management. This issue is especially critical in low-resource settings, where diagnostic tools are limited, and at-risk populations may face delayed or missed diagnoses. Since HCA typically occurs in the absence of characteristic and early clinical signs and symptoms, its prenatal diagnosis remains challenging and is generally made postnatally through placental histopathology, which is the diagnostic gold standard [6].

Recent data indicate that histologic chorioamnionitis (HCA) may be more common than clinical forms, often occurring without overt symptoms. One study found an overall incidence of approximately 16%, with similar rates observed in both term and preterm births. Interestingly, HCA was more frequently identified in cases of vaginal delivery compared to the cesarean section. In contrast, the clinical form of chorioamnionitis was diagnosed in less than 1% of cases, underscoring the subclinical nature of most HCA presentations [7].

The prevalence of chorioamnionitis in high-income countries is 1–4% of pregnancies, with both CCA and HCA implicated in 40–70% of preterm deliveries [8]. HCA has been reported in 75% of women with premature rupture of membranes (PROM) and in 30% of those without PROM [9]. The incidence is higher in preterm pregnancies compared to term pregnancies. Chorioamnionitis is associated with 40–70% of preterm births [10]. In term births, however, clinical chorioamnionitis has been diagnosed in approximately 1–3% of cases with intact membranes [11], and in 6–10% of cases with PROM [12]. Risk factors include prolonged PROM, nulliparity, labor dystocia, frequent vaginal examinations, and lower socioeconomic status [13]. Prolonged labor (≥18 h) and PROM exceeding 12 h are strongly associated with HCA [9].

Maternal complications include postpartum hemorrhage, endometritis, and sepsis, while neonatal consequences include preterm birth, respiratory distress, sepsis, and the potential development of long-term disabilities, particularly an increased risk of neurological and neuropsychiatric disorders [14]. HCA, like clinical chorioamnionitis, also increases the risk of neurodevelopmental conditions, including cerebral palsy, autism, attention-deficit/hyperactivity disorder, and intellectual disability. Therefore, effective management of chorioamnionitis requires timely diagnosis and intervention.

We report a case of HCA occurring in a pregnant woman who was admitted to our center at 40 weeks of gestation for reduced fetal movements. The admission CTG showed a “zigzag pattern” [15]. We then conducted a comprehensive review of the literature, aimed at providing insights into the diagnosis and clinical management of HCA.

## 2. Case Report

We report the case of a 29-year-old Caucasian woman with an uncomplicated pregnancy followed in our low-risk pregnancy care unit. This was her first pregnancy, with no identified obstetric risk factors. Her past medical and surgical history was unremarkable. Regarding family history, she reported diabetes mellitus but no history of hypertension, cardiovascular disease, or thromboembolic disorders. The patient first attended the obstetric department of Umberto I Hospital at 11 weeks of gestation, with a diagnosis of a first, spontaneous pregnancy. At the time of the initial visit, vital signs and clinical examinations were normal. She maintained an active lifestyle, did not smoke, followed a healthy diet, and had no underlying medical conditions.

### Pregnancy Course and Delivery

The first-trimester ultrasound and cell-free fetal DNA test did not reveal any signs of aneuploidy. The second-trimester anomaly scan performed at 20 weeks of gestation was normal. The oral glucose tolerance test at 25 weeks of gestation yielded negative results. Third-trimester ultrasound monitoring showed a physiological progression in terms of fetal growth and placental function.

At 39 weeks of gestation, the patient was admitted to our hospital due to reduced fetal movements. She refused induction of labor and was voluntarily discharged. Five days later, at 40 weeks of gestation, the patient was readmitted to the emergency department, again due to reduced fetal movements. The patient had not undergone cervico-vaginal swabs for Group B Streptococcus during the antenatal period. On admission, laboratory tests revealed a white blood cell count of 11.8 × 10^9^/L and a C-reactive protein (CRP) level of 4560 µg/L. The biophysical profile (BPP) on admission score of 6 was recorded. The BPP, which includes ultrasound monitoring of fetal movements, fetal tone, fetal breathing, and assessment of amniotic fluid volume—with or without fetal heart rate evaluation—demonstrated reduced fetal activity. Doppler ultrasound examination was normal.

Physical examination revealed normal maternal vital signs, absence of fever, no foul-smelling vaginal discharge, and no uterine tenderness. The amniotic membranes were intact, and the obstetric examination showed an unfavorable Bishop score [16]. Maternal blood pressure was 110/75 mmHg, maternal blood glucose 89 mg/dL, and the use of any form of medication by the mother, except iron and vitamins, was not reported. As shown in Figure 1, the admission cardiotocography (CTG) displayed a “zigzag pattern” [15] in the absence of uterine contractions. Given the findings, the decision was made to proceed with an emergency cesarean section.

The CTG tracing showed a baseline fetal heart rate fluctuating between approximately 130 and 170 beats per minute, with recurrent, abrupt oscillations exceeding 25 bpm in amplitude. These fluctuations were irregular in both frequency and morphology and persisted for more than one minute. No uterine contractions were observed in the lower panel, confirming that the variability was independent of contractile activity. The baseline variability appeared markedly increased, without sustained accelerations or decelerations, and the tracing exhibited an overall pattern of heightened, disorganized oscillations around the baseline.

The female infant had Apgar scores of 8 and 9, with an arterial pH of 7.27 and a venous pH of 7.36. The base deficit was −6.7 and −5.6, respectively. Birth weight was 2930 g (11th percentile), and clinical examination at birth was normal.

During the cesarean section, following amniotomy, the presence of heavily meconium-stained amniotic fluid prompted the clinician to request a culture of the amniotic fluid. In fact, the presence of meconium in the amniotic fluid reduces the antimicrobial efficacy of neutrophils and phagocytes, increasing the risk of chorioamnionitis [17].

It is acknowledged that meconium-stained amniotic fluid is no longer considered a risk factor for neonatal infection. However, in this case, the decision to perform microbiological culture of the amniotic fluid was not based solely on this finding. Rather, the culture was requested as part of a broader diagnostic approach aimed at better understanding of the underlying condition and for research purposes, given the absence of clinical signs of chorioamnionitis and the presence of abnormal cardiotocographic findings suggestive of fetal distress.

After delivery, empirical antibiotic therapy with ampicillin and gentamicin was initiated [18], and the treatment was completed at home following discharge. This treatment regimen was prescribed in accordance with ACOG guidelines on commonly used antibiotics for the treatment of suspected intra-amniotic infection [3]. In addition, the local microbiology laboratory and infectious disease specialists were consulted to confirm the appropriateness of the antibiotic choice based on local resistance patterns.

The patient was closely monitored throughout her hospital stay and remained afebrile. On admission, her blood tests were within normal limits, and inflammatory markers remained negative even at discharge. The white blood cell count in the postpartum period was 15,000/μL. Ultrasound examination at discharge was normal. The laparotomy incision was in good condition, the endometrial cavity appeared regular, and there was no evidence of intra-abdominal collections.

Culture of the amniotic fluid identified the following two pathogens: *Bacillus* species and *Staphylococcus haemolyticus*. Both organisms were sensitive to the dual antibiotic therapy administered, and no additional antibiotics were required. No infectious complications were observed in the newborn. Postnatal developmental assessments were within normal limits, and the infant’s inflammatory markers were negative. The mother provided informed consent for publication.

## 3. Discussion

Chorioamnionitis, also known as intrauterine inflammation, is typically caused by microorganisms that can enter the amniotic cavity through the following three main routes: ascending infection from the vagina and cervix, transplacental transmission, and iatrogenic introduction during invasive procedures such as amniocentesis or chorionic villus sampling. However, the most common route of intra-amniotic infection remains the ascending migration of cervico-vaginal microorganisms through the cervical canal to the decidual–chorionic interface. This leads to diffuse colonization of the endometrial–chorionic space and subsequent spread into the fetal membranes, amniotic fluid, and ultimately the fetus.

The most identified bacteria causing chorioamnionitis are *Mycoplasma* and *Ureaplasma* species, and Group B Streptococcus (GBS). Other microorganisms associated with chorioamnionitis include *Candida* species, *Escherichia coli*, and *Gardnerella vaginalis*. Approximately 30% of cases involve polymicrobial infections [19].

Regarding acute histologic chorioamnionitis (HCA), as described in the study by Duan et al. [6], it can be classified into distinct stages based on pathological assessment by identifying the degree of inflammatory response in the placental membranes. Two types of responses are recognized as follows: the maternal inflammatory response (MIR) and the fetal inflammatory response (FIR), according to established criteria.

In fact, in case of MIR, stage 1 is defined by the presence of neutrophils in the chorion or subchorionic space, while stage 2 refers to neutrophilic infiltration of the chorionic connective tissue and/or amnion or the chorionic plate, and finally stage 3 is characterized by necrotizing chorioamnionitis with degenerating neutrophils. Any evidence of vasculitis in the umbilical cord or the chorionic plate of the placenta is considered indicative of FIR. The FIR, that is the main predictor of neonatal inflammation, could be characterized by a fetal plasma concentration of interleukin-6 >11 pg/mL. FIR stage 1 includes chorionic vasculitis or umbilical phlebitis, stage 2 is marked by involvement of the umbilical vein and one or more umbilical arteries, and stage 3 refers to necrotizing funisitis. To describe the severity of HCA, MIR/FIR stage 1 is topically classified as mild HCA, while MIR/FIR stages 2–3 are classified as moderate-to-severe HCA [6].

As previously reported, chorioamnionitis is associated with both maternal and neonatal complications. In particular, as shown by Venkatesh et al., chorioamnionitis is associated with approximately 2- to 3.5-fold increased odds of adverse neonatal outcomes, including perinatal death, early-onset neonatal sepsis, septic shock, pneumonia, meningitis, intraventricular hemorrhage (IVH), cerebral white matter damage, and long-term disabilities such as cerebral palsy, retinopathy of prematurity, necrotizing enterocolitis (NEC), as well as morbidity related to preterm birth [20].

Chorioamnionitis is in fact a well-known non-hypoxic cause of fetal impairment. The inflammatory pathway leading to neurological injury is more common than the hypoxic pathway, although it is frequently misdiagnosed or underdiagnosed. Histologic chorioamnionitis (HCA) lacks specific diagnostic hallmarks, making interpretation challenging. In this context, cardiotocography (CTG) plays a crucial role in evaluating fetal oxygenation status and remains the gold standard for detecting fetal hypoxia. However, no specific CTG features have yet been definitively linked to the diagnosis of either clinical or subclinical chorioamnionitis. Notably, the first study investigating CTG changes in both subclinical and clinical chorioamnionitis was published by Galli et al. as recently as 2019 [21].

In the present case, the patient did not exhibit any classical clinical signs or symptoms of chorioamnionitis (such as maternal fever or fetal/maternal tachycardia), except for heavily meconium-stained and foul-smelling amniotic fluid. The decision to perform an emergency cesarean section was based on the detection of a ‘zigzag pattern’ [15] on the CTG tracing—highly suggestive of possible fetal compromise—combined with reports of reduced fetal movements. The diagnostic value of the “zigzag pattern” (ZZP) is not yet universally accepted, and further large-scale validation studies and the definition of standardized interpretation criteria are needed. Although promising, the ZZP should be interpreted cautiously and always within the context of the overall clinical picture.

The presence of baseline fetal heart rate variability (FHRV) reflects the integrity of the central nervous system and indicates the fetus’s capacity to adapt to intrauterine conditions. If the fetus is exposed too rapidly to hypoxic stress, resulting in evolving hypoxia, the simultaneous increase in sympathetic nervous system activity (to enhance oxygen delivery) and parasympathetic activity (to reduce myocardial workload) may lead to autonomic instability [22,23].

This autonomic instability can be observed on cardiotocographic (CTG) monitoring as rapid, irregular, and abrupt fluctuations in the baseline fetal heart rate, with an amplitude of >25 beats per minute. The authors define this as the “zigzag pattern” when it persists for at least one minute [15,24].

The main difference between the “skipping pattern” (SP) and the “zigzag pattern” (ZZP) lies in uniformity and duration. The SP is typically sustained for at least 30 min. A retrospective study published in 2019 evaluated the incidence of SP and ZZP during labor and correlated them with perinatal outcomes, with specific attention to the duration of ZZP. The study found that infants who exhibited a ZZP during active maternal pushing had statistically significant lower Apgar scores at 1 and 5 min, higher rates of moderate and mild umbilical artery acidosis, and an 8.7- to 11.4-fold increased risk of neonatal hospitalization [15].

In 2024, E. Chandraharan proposed the “Chorio Duck Score,” an evidence-based scoring system based on CTG features associated with subclinical and clinical chorioamnionitis. The score aims to support early diagnosis of fetal inflammatory response syndrome (FIRS) [17]. The primary goal is to avoid hypoxic stress during labor in the presence of underlying fetal inflammation, thereby preventing fetal neurological damage.

The emphasis is on recognizing chorioamnionitis as a potentially fatal fetal condition that presents with specific signs and should be managed proactively—before the appearance of maternal signs such as pyrexia and tachycardia. In fact, changes in maternal clinical parameters may not occur until the later stages of disease progression [17,25].

Although the Chorio Duck Score represents a promising tool for assessing chorioamnionitis, its clinical adoption remains limited due to incomplete validation and practical barriers to implementation in routine obstetric practice.

Pereira and Chandraharan proposed a “Fetal Monitoring Checklist” to be used at the time of admission in labor, in order to exclude pre-existing causes of fetal compromise [25] and to assess whether the fetus is fit to withstand the progressive hypoxic stress of labor.

Physiologically, during labor—and especially during uterine contractions—uteroplacental perfusion is reduced, and blood flow to the intervillous space is intermittently interrupted, exposing the fetus to hypoxic stress [26]. Fetuses affected by chorioamnionitis exhibit a higher metabolic rate and increased oxygen demand; when exposed to prolonged hypoxic stress, the risk of anaerobic metabolism increases, as does the risk of neurological damage due to elevated lactate levels. In particular, elevated fetal/neonatal levels of pro-inflammatory cytokines and chemokines—especially the tumor necrosis factor—can lead to fetal or neonatal brain injury, resulting in ischemia, intraventricular hemorrhage (IVH), and periventricular leukomalacia [27].

In our case, abnormalities were detected in the admission cardiotocography, showing features suggestive of chorioamnionitis at the onset of labor. As a result, delivery was expedited via emergency cesarean section.

Considering the current diagnostic challenges associated with histologic chorioamnionitis (HCA), as highlighted in this case report, there is a clear need for further research aimed at identifying early and specific markers of subclinical intrauterine inflammation. In particular, the role of cardiotocographic findings—especially the so-called “zigzag pattern”—as a potential early indicator of fetal compromise in the absence of maternal clinical signs represents a promising yet under-investigated area. Prospective, large-scale studies are essential to validate this pattern as a reliable diagnostic tool in clinical practice, ultimately improving early detection of at-risk pregnancies and reducing the incidence of long-term neonatal complications.

## 4. Conclusions

Maternal monitoring of active fetal movements is a key element in the assessment of fetal well-being at term and is a simple, reproducible, and reliable method. At term, in cases of reduced fetal movements, regardless of the presence of a cardiotocographic tracing suggestive of possible fetal compromise, delivery is recommended. Alternatively, if the patient refuses induction of labor, periodic cardiotocographic monitoring and assessment of the fetal biophysical profile are recommended.

In conclusion, histologic chorioamnionitis (HCA), like clinical chorioamnionitis, increases the risk of neurodevelopmental disorders, including cerebral palsy, autism spectrum disorder, attention-deficit/hyperactivity disorder (ADHD), and intellectual disability. These associations are more pronounced in preterm infants compared to those born at term [28]. Therefore, early identification and timely intervention offer substantial benefits in reducing the burden of neurological complications in affected infants.

Unfortunately, there is still a lack of useful signs and symptoms for the early diagnosis of this condition. Early identification of the “zigzag pattern” at cardiotocographic monitoring, in the absence of other signs and/or symptoms, shows promising potential in the timely diagnosis of this condition. Its role should be further investigated in future research in order to effectively reduce the risk of neonatal complications.

## Figures and Tables

**Figure 1 healthcare-13-01726-f001:**
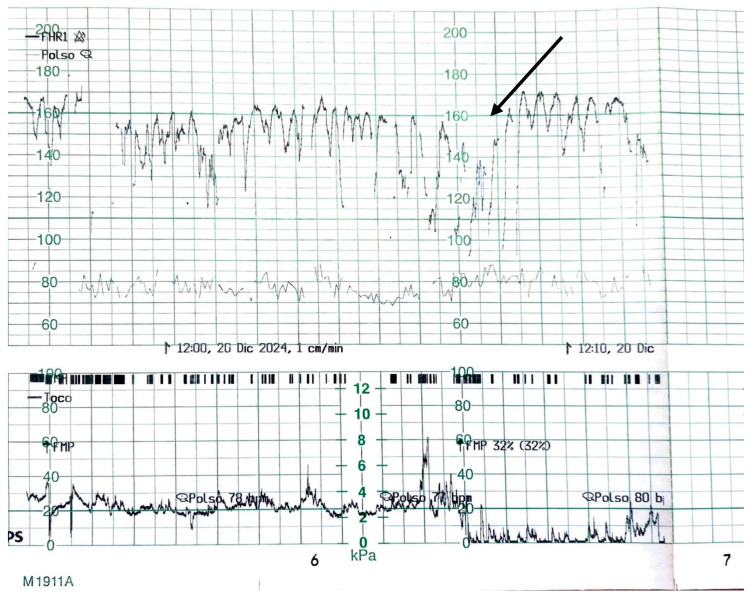
Admission CTG before the emergency cesarean section. The arrow in the figure highlights the characteristic ZigZag pattern.

## Data Availability

Available from the authors at reasonable request.

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
