# Peer review of "Zigzag Fetal Heart Rate Pattern in an Uncomplicated Pregnancy with Dual Intrauterine Infection Detected During Labor with Intact Membranes: A Case Report"

_healthcare, 2025, doi:10.3390/healthcare13141726_

Round 1

Reviewer 1 Report

Comments and Suggestions for Authors

It is not presented the score of Biophysical profile

It must be presented the result of anatomopathological exam of the placenta and membranes

Author Response

  1. It is not presented the score of Biophysical profile

The biophysical profile score has been added.

  1. It must be presented the result of anatomopathological exam of the placenta and membranes

Unfortunately, since the full placenta and membranes were sent to microbiological lab no anatomopathological data were available.

Reviewer 2 Report

Comments and Suggestions for Authors

Very well constructed and presented manuscript explaining an important and often unsuspected complication of pregnancy affecting both maternal and fetal outcomes. The subtle nature but potentially profound impact is well explained. The case study approach makes for an interesting read. I definitely think this is a worthwhile manuscript and needs to be published. The follow-up implications for further research are the only area where I think the authors could expand the discussion.

Author Response

  1. Very well constructed and presented manuscript explaning an important and often unsuspected complication of pregnancy affecting both maternal and fetal outcomes. The subtle nature but potentially profound impact is well explained. The case study approach makes for an interesting read. I definitely think this is a worthwhile manuscript and needs to be published. The follow-up implications for further research are the only area where I think the authors could expand the discussion.

Thanks for your positive comments. The discussion on implications for future research studies has been expanded.

Reviewer 3 Report

Comments and Suggestions for Authors

This is an interesting article but it contains some elements that can create some debates.

For example:

"Culture examination of amniotic fluid identified two pathogens: Bacillus species and Staphylococcus  haemolyticus."  What kind of Bacillus? It is not enough to mention Bacillus spp. You have to show the exact type of bacillus.

"During the caesarean section following  amniorexis the finding of heavily meconium-stained fluid led the clinician to the decision to request culture of the amniotic fluid."

Nowadays, amniotic stained fluid it is not considered a risk factor for infection in neonates. This type of approach was used some years ago, but not today.

Also, you did not offer enough arguments for building a case around the presence of the 2 bacteria you mentioned. In my opinion, you should add more lab data and add more arguments around the main topic of your article. Otherwise, this will not be a well argued thesis.

Also, the lab values you mentioned are not necessarily specific for CA. 

Please add more data about the newborn and the evolution during the maternity stay. Also add more info about the newborn's lab data.

Author Response

This is an interesting article but it contains some elements that can create some debates.

  1. For example: "Culture examination of amniotic fluid identified two pathogens: Bacillus species and Staphylococcus haemolyticus." What kind of Bacillus? It is not enough to mention Bacillus spp. You have to show the exact type of bacillus.

Unfortunately, the microbiological examination of the amniotic fluid identified two pathogens: Staphylococcus haemolyticus and Bacillus species, without specifying the exact type of Bacillus.

  1. "During the caesarean section following amniorexis the finding of heavily meconium-stained fluid led the clinician to the decision to request culture of the amniotic fluid." Nowadays, amniotic stained fluid it is not considered a risk factor for infection in neonates. This type of approach was used some years ago, but not today.

It is acknowledged that meconium-stained amniotic fluid is no longer considered a standalone risk factor for neonatal infection. However, in this case, the decision to perform microbiological culture of the amniotic fluid was not made solely based on this finding. Rather, the culture was requested as part of a broader diagnostic approach aimed at better understanding the underlying condition and for research purposes, given the absence of clinical signs of chorioamnionitis and the presence of abnormal cardiotocographic findings suggestive of fetal distress.

  1. Also, you did not offer enough arguments for building a case around the presence of the 2 bacteria you mentioned. In my opinion, you should add more lab data and add more arguments around the main topic of your article. Otherwise, this will not be a well argued thesis. Also, the lab values you mentioned are not necessarily specific for CA.

More detailed laboratory data have been added in an effort to strengthen the discussion around the main topic.

  1. Please add more data about the newborn and the evolution during the maternity stay. Also add more info about the newborn's lab data.

Based on the available data, additional information on the newborn and the clinical course during the hospital stay has been added.

Reviewer 4 Report

Comments and Suggestions for Authors

Introduction (Page 1, Lines 30–64)

The introduction outlines the clinical context well, but would benefit from expanding the epidemiological perspective. Consider including global and regional prevalence data on histologic chorioamnionitis (HCA) and associated neonatal outcomes.

Line 43: The distinction between clinical and histologic chorioamnionitis is appropriate; however, the diagnostic criteria and relevance of distinguishing both types in a public health framework should be emphasized.

Lines 63–64: Statements regarding timely diagnosis are general; referencing diagnostic challenges in low-resource settings or among at-risk populations would strengthen the manuscript's applicability.

Case Description (Pages 2–3, Lines 69–120)

Line 76: The manuscript lacks pre-delivery infection screening data (e.g., Group B Streptococcus culture, CRP levels, or CBC). The inclusion of these standard prenatal indicators would add clinical depth.

Lines 97–101: The description of the ZigZag CTG pattern should be more quantitatively detailed. Please include exact definitions, such as amplitude thresholds and duration criteria.

Lines 107–110: The rationale behind empiric antibiotic selection (ampicillin and gentamicin) is not discussed in the context of standard clinical guidelines (e.g., ACOG, WHO). Please provide justification and cite protocols.

Results (Pages 3–4, Lines 102–120)

The reporting of clinical outcomes is clear. However, the manuscript does not mention any planned postnatal developmental assessments, which are critical in evaluating long-term sequelae of intrauterine infection.

Line 121: Please clarify whether the neonate received additional monitoring (e.g., blood cultures, inflammatory markers). These data would enhance clinical completeness.

Discussion (Pages 4–6, Lines 123–218)

The discussion appropriately references relevant literature but requires a greater critical analysis of the current case compared to existing case reports or cohort studies.

Lines 160–168: The diagnostic value of the 'ZigZag pattern' is not universally accepted. The manuscript could discuss the lack of validation and limitations of this CTG feature in large-scale clinical practice.

Lines 172–174: The assertion that the absence of FIR led to a favorable neonatal outcome is speculative without corroborating biomarkers. Please state this more cautiously as a hypothesis.

Lines 195–206: The 'Chorio Duck Score' is a promising tool, but its clinical adoption is limited. A brief critique of its validation status or implementation barriers would be appropriate.

Conclusion (Page 6, Lines 219–229)

The conclusion restates findings but does not offer clear clinical recommendations. Please consider adding:

    • Proposed indications for CTG-based early delivery.
    • Suggestions for surveillance in low-risk term pregnancies with reduced fetal movement.Line 228: The potential of CTG in early HCA diagnosis is promising, but generalization could be made cautiously and framed as a hypothesis for future research, not a clinical standard.

Figures and Tables (Page 3, Figure 1): The CTG trace is under-annotated and lacks interpretation guidance. Please add scale, time axis, and arrows to mark the ZigZag pattern. If possible, include a figure legend explaining its diagnostic value and criteria.

 Language and Clarity

The manuscript contains numerous grammatical and syntactical issues. Examples:

    • "Actually showed reduced fetal movements", "Demonstrated reduced fetal movements"
    • “Blunts the antibacterial action…”- “Reduces the antimicrobial efficacy…”
Comments on the Quality of English Language

The manuscript requires careful English language revision to improve clarity, consistency, and professionalism. Several areas contain grammatical inaccuracies, informal expressions, and inconsistent tense usage that affect the overall readability and scientific tone of the paper.

For example:

Line 92: “actually showed reduced fetal movements” – the word “actually” is unnecessary and informal. A clearer phrasing would be: “demonstrated reduced fetal movements.”

Line 107: “blunts the antibacterial action of the neutrophils” – the verb “blunts” is too informal in a scientific context; consider revising to “reduces the antimicrobial efficacy of neutrophils.”

Line 105: The phrase “led the clinician to the decision to request culture” is overly wordy. A more concise version would be: “prompted the clinician to request a culture.”

Additionally, some sentences are too long or redundant, and punctuation is occasionally inconsistent.

Author Response

Introduction (Page 1, Lines 30–64)

  1. The introduction outlines the clinical context well, but would benefit from expanding the epidemiological perspective. Consider including global and regional prevalence data on histologic chorioamnionitis (HCA) and associated neonatal outcomes.

Data on prevalence and associated neonatal outcomes have been expanded.

  1. Line 43: The distinction between clinical and histologic chorioamnionitis is appropriate; however, the diagnostic criteria and relevance of distinguishing both types in a public health framework should be emphasized.

The text has been enriched based on the reviewer’s suggestions.

  1. Lines 63–64: Statements regarding timely diagnosis are general; referencing diagnostic challenges in low-resource settings or among at-risk populations would strengthen the manuscript's applicability.

The text has been enriched based on the reviewer’s suggestions.

Case Description (Pages 2–3, Lines 69–120)

  1. Line 76: The manuscript lacks pre-delivery infection screening data (e.g., Group B Streptococcus culture, CRP levels, or CBC). The inclusion of these standard prenatal indicators would add clinical depth.

These standard prenatal indicators have been added to the case description.

  1. Lines 97–101: The description of the ZigZag CTG pattern should be more quantitatively detailed. Please include exact definitions, such as amplitude thresholds and duration criteria.

The description of the ZigZag CTG pattern has been made more detailed.

  1. Lines 107–110: The rationale behind empiric antibiotic selection (ampicillin and gentamicin) is not discussed in the context of standard clinical guidelines (e.g., ACOG, WHO). Please provide justification and cite protocols.

The choice of antibiotics has been justified in relation to the guidelines.

Results (Pages 3–4, Lines 102–120)

  1. The reporting of clinical outcomes is clear. However, the manuscript does not mention any planned postnatal developmental assessments, which are critical in evaluating long-term sequelae of intrauterine infection.

We appreciate the reviewer’s valuable comment regarding the importance of postnatal developmental assessments in evaluating long-term outcomes following intrauterine infection. At the time of writing, however, only a few months have passed since the delivery, which is insufficient to conduct meaningful neurodevelopmental assessments. Nonetheless, the infant is currently undergoing regular pediatric follow-up, and long-term evaluations are planned in accordance with clinical guidelines. We intend to report these outcomes in future updates or follow-up studies.

  1. Line 121: Please clarify whether the neonate received additional monitoring (e.g., blood cultures, inflammatory markers). These data would enhance clinical completeness.

The available postnatal assessments have been included.

Discussion (Pages 4–6, Lines 123–218)

The discussion appropriately references relevant literature but requires a greater critical analysis of the current case compared to existing case reports or cohort studies.

  1. Lines 160–168: The diagnostic value of the 'ZigZag pattern' is not universally accepted. The manuscript could discuss the lack of validation and limitations of this CTG feature in large-scale clinical practice.

We acknowledge that the diagnostic value of the ZigZag pattern (ZZP) is not yet universally accepted. We have added a brief discussion in the manuscript about the need for further large-scale validation and the limitations of using ZZP in clinical practice.

  1. Lines 172–174: The assertion that the absence of FIR led to a favorable neonatal outcome is speculative without corroborating biomarkers. Please state this more cautiously as a hypothesis.

We agree that the assertion regarding the absence of fetal inflammatory response (FIR) leading to a favorable neonatal outcome is speculative without supporting biomarker data. Accordingly, we have revised the manuscript.

  1. Lines 195–206: The 'Chorio Duck Score' is a promising tool, but its clinical adoption is limited. A brief critique of its validation status or implementation barriers would be appropriate.

We acknowledge that although the Chorio Duck Score is a promising tool, its clinical adoption remains limited. In response, we have added a brief discussion in the manuscript addressing the current status of its validation and the barriers to its widespread implementation in clinical practice.

Conclusion (Page 6, Lines 219–229)

  1. The conclusion restates findings but does not offer clear clinical recommendations. Please consider adding:

Proposed indications for CTG-based early delivery.

Suggestions for surveillance in low-risk term pregnancies with reduced fetal movement.

The clinical recommendations suggested by the reviewer have been included.

  1. Line 228: The potential of CTG in early HCA diagnosis is promising, but generalization could be made cautiously and framed as a hypothesis for future research, not a clinical standard.

The reviewer’s suggestion was followed by making corrections to the text.

Figures and Tables (Page 3, Figure 1):

  1. The CTG trace is under-annotated and lacks interpretation guidance. Please add scale, time axis, and arrows to mark the ZigZag pattern. If possible, include a figure legend explaining its diagnostic value and criteria.

The CTG trace includes a scale and a time axis, and an arrow highlighting the ZigZag pattern has been added as suggested.

Comments on the Quality of English Language

  1. The manuscript requires careful English language revision to improve clarity, consistency, and professionalism. Several areas contain grammatical inaccuracies, informal expressions, and inconsistent tense usage that affect the overall readability and scientific tone of the paper.

For example:

Line 92: “actually showed reduced fetal movements” – the word “actually” is unnecessary and informal. A clearer phrasing would be: “demonstrated reduced fetal movements.”

Line 107: “blunts the antibacterial action of the neutrophils” – the verb “blunts” is too informal in a scientific context; consider revising to “reduces the antimicrobial efficacy of neutrophils.”

Line 105: The phrase “led the clinician to the decision to request culture” is overly wordy. A more concise version would be: “prompted the clinician to request a culture.” Additionally, some sentences are too long or redundant, and punctuation is occasionally inconsistent.

An English language revision has been carried out.

Round 2

Reviewer 3 Report

Comments and Suggestions for Authors

Thank you for adding more lab data and more information about the labour and the evolution of the newborn.

About the altered CTG you should consider offering info about the mother's blood pressure and glycemia. Both of them could alter CTG. Also, have you checked if the pregnant woman is undergoing any kind of medical treatment? Maybe some drugs could alter CTG. Please address these possible problems.

Even though you added more info I am still not enlightened about the conclusion you want to draw with this article...maybe you should change the title to be more connected with the article's body.

Author Response

Thank you for adding more lab data and more information about the labour and the evolution of the newborn.

About the altered CTG you should consider offering info about the mother's blood pressure and glycemia. Both of them could alter CTG. Also, have you checked if the pregnant woman is undergoing any kind of medical treatment? Maybe some drugs could alter CTG. Please address these possible problems.

Blood pressure were normal as well glycemia, no medical treatment were used

Even though you added more info I am still not enlightened about the conclusion you want to draw with this article...maybe you should change the title to be more connected with the article's body.

Title was changed as suggested